# Imaging the state-to-state charge-transfer dynamics between the spin-orbit excited Ar$^+$($^2$P$_{1/2}$) ion and N$_2$

Guodong Zhang [1,2,4], Dandan Lu[3,4], Hua Guo [3] ✉ & Hong Gao [1,2] ✉

Ar$^+$+N$_2$ → Ar+N$_2^+$ has served as a paradigm for charge-transfer dynamics studies during the last several decades. Despite significant experimental and theoretical efforts on this model system, state-resolved experimental investigations on the microscopic charge-transfer mechanism between the spin-orbit excited Ar$^+$($^2$P$_{1/2}$) ion and N$_2$ have been rare. Here, we measure the first quantum state-to-state differential cross sections for Ar$^+$+N$_2$ → Ar+N$_2^+$ with the Ar$^+$ ion prepared exclusively in the spin-orbit excited state $^2$P$_{1/2}$ on a crossed-beam setup with three-dimensional velocity-map imaging. Trajectory surface-hopping calculations qualitatively reproduce the vibrationally dependent rotational and angular distributions of the N$_2^+$ product. Both the scattering images and theoretical calculations show that the charge-transfer dynamics of the spin-orbit excited Ar$^+$($^2$P$_{1/2}$) ion differs significantly from that of the spin-orbit ground Ar$^+$($^2$P$_{3/2}$) when colliding with N$_2$. Such state-to-state information makes quantitative understanding of this benchmark charge-transfer reaction within reach.

Ion-molecule reactions are ubiquitous and play fundamental roles in many gaseous environments such as interstellar media[1,2], planetary atmospheres[3,4], and plasmas[5]. An in-depth understanding of the microscopic mechanisms for all the relevant ion-molecule reactions is pivotal for modeling these environments. The charge-transfer reaction Ar$^+$+N$_2$ → Ar+N$_2^+$ has long served as a model system for studying gas-phase ion-molecule reaction dynamics and has been subjected to extensive experimental and theoretical investigations over the last half century[6-16]. However, many discrepancies remain between different experiments, and between experimental measurements and theoretical calculations, which prevented us from reaching a definitive understanding of the microscopic dynamics of this prototypical charge-transfer reaction.

Although many different experimental techniques have been developed for studying gas-phase ion-molecule reaction dynamics[17], the most insightful way remains the crossed-beam approach under single collision conditions[18,19]. The full mapping of the kinetic energy and angular distributions of the scattering products in crossed-beam experiments provides the most detailed information about the potential energy surface (PES) of the reactive system. The first crossed-beam experiment on the charge-transfer reaction Ar$^+$+N$_2$ → Ar+N$_2^+$ was reported by Futrell and coworkers in the early 1980s on a conventional crossed-beam setup with a rotatable product detector[6]. The N$_2^+$ product was found to be populated predominantly in the $v'$ = 1 vibrational level and mainly scattered into the forward direction at the collision energies of 1.73 and 4.01 eV, indicating a direct electron hopping mechanism with negligible momentum transfer. However, later experiments from the same laboratory at the collision energies of ~1 eV found surprisingly that the N$_2^+$ product could be populated in all energetically accessible vibrational levels and each of these vibrational levels was scattered into a different angular region[7,8]. This unexpected finding has puzzled researchers for many years, as it has never been reproduced by theoretical calculations[10,12,13].

[1]Beijing National Laboratory for Molecular Sciences (BNLMS), Institute of Chemistry, Chinese Academy of Sciences, 100190 Beijing, China. [2]University of Chinese Academy of Sciences, 100049 Beijing, China. [3]Department of Chemistry and Chemical Biology, Center for Computational Chemistry, University of New Mexico, Albuquerque, NM 87131, USA. [4]These authors contributed equally: Guodong Zhang, Dandan Lu. ✉e-mail: hguo@unm.edu; hong-gao2017@iccas.ac.cn

The application of the velocity-map imaging (VMI) technique in crossed-beam experiments has significantly advanced ion-molecule scattering dynamics studies during the last two decades[20–24]. In 2006, Wester and coworkers applied their first-generation VMI-based crossed-beam setup to image the charge-transfer dynamics between $Ar^+$ and $N_2$. Significant vibrational excitation of $N_2^+$ was implicated, but the limited energy resolution prevented them from reaching a definitive conclusion[14]. In 2013, they revisited the same charge-transfer reaction with their second-generation VMI-based crossed-beam setup, with much-improved ion beam quality and imaging resolution[15]. The obtained product images showed that the $v' = 1$ vibrational level of $N_2^+$ only dominates in the forward scattering direction, while higher vibrational excitation (up to $v' = 6$) becomes more important in larger scattering angles. As the collision energy decreases, scattering into larger angles becomes more important for all energetically accessible product vibrational states. The observed vibrationally dependent product angular distributions qualitatively agreed with the calculated results based on a semiclassical Landau-Zener model[12,13]. However, the product vibrational branching ratio $N_2^+(v' = 1)/N_2^+(v' = 2)$ deduced from the imaging was significantly smaller than both the theoretical prediction[12] and the measured value of the quantum state selected guided ion beam experiment[9]. This was attributed to the coexistence of both the spin-orbit ground $Ar^+(^2P_{3/2})$ and excited $Ar^+(^2P_{1/2})$ in their ion beam, and the charge transfer between the spin-orbit excited $Ar^+(^2P_{1/2})$ and $N_2$ produces $N_2^+$ in higher vibrational levels[9,16].

To distinguish the two spin-orbit levels of the $Ar^+$ ion and to gain a deeper insight into the charge-transfer dynamics, we have recently constructed a new three-dimensional VMI-based ion-molecule crossed-beam setup with a pulsed photoionization-based quantum state selected ion beam source[25,26]. We reported the first quantum state-to-state charge-transfer dynamics study on the reaction $Ar^+ + N_2 \rightarrow Ar + N_2^+$ with the $Ar^+$ ion prepared exclusively in the spin-orbit ground state $^2P_{3/2}$ by using the resonance-enhanced multiphoton ionization (REMPI) method[27]. The optimized imaging resolution allowed us to resolve the individual vibrational levels of the $N_2^+$ product in the forward scattering region. Product signals scattered into large scattering angles as reported by Wester and coworkers[15] were also observed in our work and were attributed to rotationally excited instead of vibrationally excited $N_2^+$ products, which was confirmed by trajectory surface-hopping calculations. The product vibrational state-specific charge-transfer mechanisms of this model system have been clearly elucidated for the first time.

As discussed above, the charge transfer between the spin-orbit excited $Ar^+(^2P_{1/2})$ ion and $N_2$ could potentially contribute to the observed results in most of the previous scattering experiments, and its effect could not be ignored when explaining the experimental results. Previous experimental and theoretical studies have suggested that the outcome from the charge-transfer reaction of $Ar^+(^2P_{1/2}) + N_2$ differs significantly from that of $Ar^+(^2P_{3/2}) + N_2$[9,12,13,16]. Despite many scattering experiments focusing on the spin-orbit ground $Ar^+(^2P_{3/2})$ ion[6–8,14,15], quantum state-to-state differential cross sections (DCSs) for the charge-transfer dynamics between the spin-orbit excited $Ar^+(^2P_{1/2})$ ion and $N_2$ have not been fully explored before. In this study, we report high-resolution scattering images of the charge-transfer reaction between the spin-orbit excited $Ar^+(^2P_{1/2})$ ion and $N_2$ at several center-of-mass (COM) collision energies, which reveal quite different features from those between the spin-orbit ground $Ar^+(^2P_{3/2})$ ion and $N_2$ as reported recently[27]. Trajectory surface-hopping calculations are performed, which qualitatively capture the main features of the scattering dynamics. The synergistic experimental and theoretical study provides deeper insights into the microscopic charge-transfer mechanism between the spin-orbit excited $Ar^+(^2P_{1/2})$ ion and $N_2$ for the first time.

## Results

Three-dimensional velocity distributions of the $N_2^+$ product for the spin-orbit state selected charge-transfer reaction $Ar^+(^2P_{1/2}) + N_2(X ^1\Sigma_g^+, v = 0) \rightarrow Ar + N_2^+(v', J')$ have been measured at four COM collision energies, namely 1.58, 1.10, 0.83 and 0.57 eV. In Fig. 1a, the central slice image cut from the experimentally measured three-dimensional velocity distribution at 1.58 eV is presented, and the red concentric rings labeled with numbers represent the kinematic cutoffs for the vibrational levels of the $N_2^+$ product. The central slice images at the other three COM collision energies are presented in Supplementary Fig. 1. To gain a more quantitative view of the $N_2^+$ product velocity

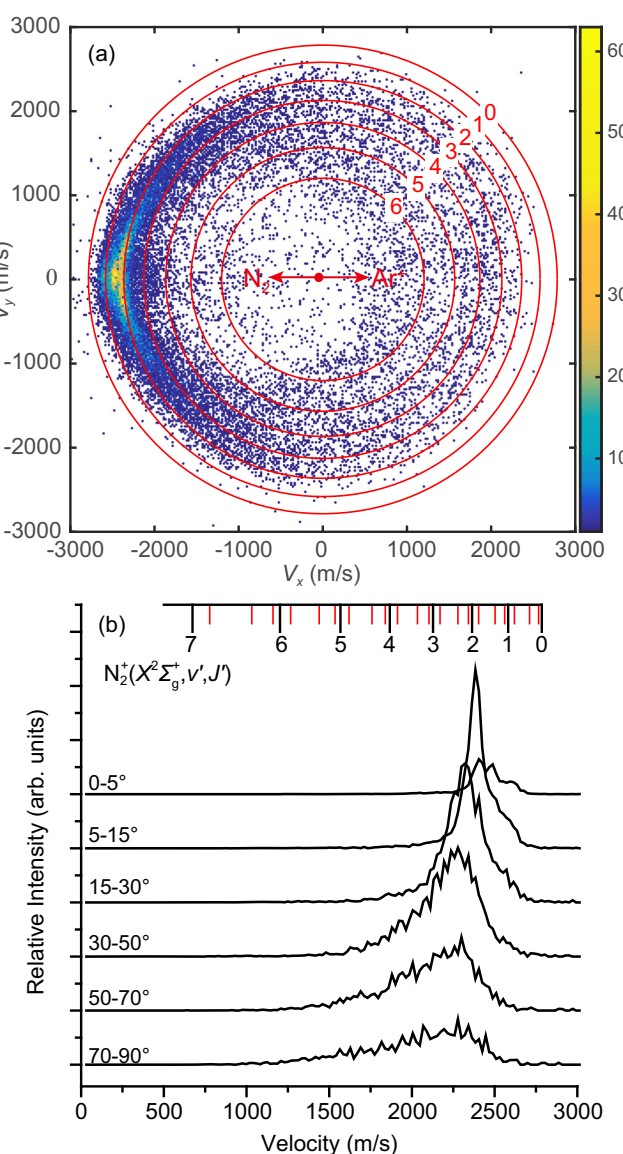

**Fig. 1 | Product imaging and integrated speed distributions. a** The central slice image of the three-dimensional $N_2^+$ velocity distribution for the charge-transfer process between the spin-orbit excited $Ar^+(^2P_{1/2})$ ion and $N_2$ at the center-of-mass (COM) collision energy of 1.58 eV. The moving directions of $N_2$ and $Ar^+$ beams in the COM frame are indicated by the red arrows, and the kinematic cutoffs for each vibrational level of $N_2^+$ considering the anharmonic corrections are indicated by the red concentric rings. The color bar represents the absolute product ion count. **b** The $N_2^+$ product velocity distributions in various angular ranges. The vibrational levels of the $N_2^+$ product are indicated by the black droplines, and the positions of the rotational levels $J' = 10, 20,$ and $30$ of $N_2^+$ in each vibrational level are indicated by the red droplines.

distribution, we integrated the image in various scattering angular ranges from 0 to 90°, and the resulting velocity distributions are presented in Fig. 1b. It can be noticed immediately from Fig. 1 that the detailed scattering features for the spin-orbit excited $Ar^+(^2P_{1/2})$ ion are profoundly different from those for the spin-orbit ground $Ar^+(^2P_{3/2})$ ion as reported recently[27]. The $v' = 1$ and $v' = 2$ vibrational levels of the $N_2^+$ products are only partially resolved from each other in the forward scattering direction for $Ar^+(^2P_{1/2})$ as shown in Fig. 1a. The $v' = 1$ level appears as a reproducible shoulder adjacent to the $v' = 2$ level in the integrated $N_2^+$ product velocity distributions in the angular ranges from 0 to 30° as shown in Fig. 1b. While for $Ar^+(^2P_{3/2})$ at the same COM collision energy, the $v' = 1$ and $v' = 2$ levels can be clearly resolved in the forward direction, see Fig. 1 in ref. 27. Despite the incompletely resolved $v' = 1$ and $v' = 2$ vibrational levels in the forward scattering region, the $v'=2$ level has an unmistakably higher population than that of the $v' = 1$ level, as indicated clearly by the integrated velocity distributions in Fig. 1b. For $Ar^+(^2P_{3/2})$, our recent study has established that the $v'=1$ level dominates the $N_2^+$ product in almost all scattering angles[27]. This observation qualitatively agrees with the previous quantum state selected guided ion beam experiment[9] and theoretical calculations based on the Landau-Zener model[12,13]. Both scattering images of $Ar^+(^2P_{3/2})$ and $Ar^+(^2P_{1/2})$ show detectable signal intensities in the backward scattering regions, but the backward scattered $N_2^+$ from $Ar^+(^2P_{1/2})$ is populated at much higher vibrational levels (up to $v' = 6$) than that from $Ar^+(^2P_{3/2})$, as shown in Fig. 1a. Finally, the $N_2^+$ product from the charge-transfer reaction with $Ar^+(^2P_{3/2})$ is much more forward peaked than that from the reaction with $Ar^+(^2P_{1/2})$, which will be discussed further below. Despite the many differences as discussed above, common features between the scattering images of $Ar^+(^2P_{3/2})$ and $Ar^+(^2P_{1/2})$ should also be noticed. For example, the velocity distribution peak moves toward lower speed values at larger scattering angular ranges for both $Ar^+(^2P_{3/2})$ and $Ar^+(^2P_{1/2})$, indicating larger internal excitation of the product ions. On the other hand, the main peak does not exceed $v' = 3$, as shown in Fig. 1b. This indicates that the internal excitation can only be attributed to the rotational degree of freedom of the $N_2^+$ product.

Due to the partially resolved product vibrational levels in the scattering images and the vibrationally dependent rotational and angular distributions, as shown later, a quantitative determination of the product vibrational population is not feasible from the scattering image. Hence, only an estimation of the product vibrational population is given and presented in Fig. 2, which is obtained by using a Gaussian profile fitting process similar to that by Wester and coworkers (see Supplementary Note 2 for details)[15]. The fitting error is much smaller than 1%, thus not shown in Fig. 2. As shown in Fig. 2, the $v' = 2$ level dominates at all the four COM collision energies studied here with its population increasing from ~50% to ~80% as the collision energy decreases from 1.58 eV to 0.57 eV. Besides the $v' = 2$ levels, significant populations in the $v' = 1$ and $v' > 2$ (up to $v' = 6$ at 1.58 eV) vibrational levels are also observed. This qualitatively agrees with several previous experimental and theoretical studies[9,12,13]. In the quantum state selected guided ion beam experiment by Ng and coworkers[9], no other vibrational levels than $v'=2$ were observed at the collision energy of 1.2 eV. In the theoretical calculation by Candori et al. based on the Landau-Zener model[12,13], the $v' = 2$ level was found to dominate the charge-transfer process up to the collision energy of ~3 eV, and significant populations to vibrational levels of $v' = 3-6$ were also predicted whenever they are energetically accessible. The population of the $v' = 1$ level was predicted to be low at 1.58 eV, while gradually becoming more significant as the collision energy decreased, which qualitatively agrees with the current experiment. The trajectory surface-hopping calculation is performed in the current study (see "Methods") and the calculated vibrational populations of $N_2^+$ are presented in Fig. 2. The trajectory surface-hopping calculation did predict much higher populations in the $v' = 2$ and 3 levels for the charge-

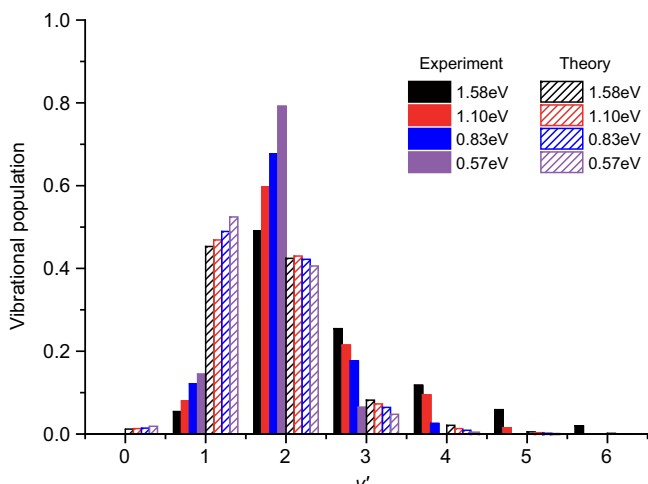

**Fig. 2 | Product vibrational populations.** Comparison of experimentally estimated vibrational populations (the filled histograms) of the $N_2^+$ product for the charge-transfer reaction $Ar^+(^2P_{1/2}) + N_2 \rightarrow Ar + N_2^+(v', J')$ with the corresponding results calculated by the trajectory surface-hopping method (the shaded histograms). The black, red, blue, and purple histograms represent the $N_2^+$ product vibrational populations at the COM collision energies of 1.58 eV, 1.10 eV, 0.83 eV, and 0.57 eV, respectively.

transfer reaction with the spin-orbit excited $Ar^+(^2P_{1/2})$ ion than those with the ground $Ar^+(^2P_{3/2})$ ion. However, the calculation overestimated the population of the $v' = 1$ level and underestimated that of the $v' = 2$ level when compared with the experimental measurements.

The angular distributions of the $N_2^+$ products in the $v' = 1$ and 2 levels at 1.58 eV are deduced from the scattering image and presented in Fig. 3a. The corresponding product angular distributions calculated by the trajectory surface-hopping method are shown in Fig. 3b. Those at the other three collision energies are shown in Supplementary Fig. 2. At first glance, both experimental measurements and theoretical calculations show that the $N_2^+$ products are strongly forward peaked with the $v' = 2$ level scattered into slightly larger angular ranges than the $v' = 1$ level. Quantitatively, however, the experimental measurements show that the $N_2^+$ products are scattered into much larger angular ranges than predicted by the theoretical calculations. The $v' = 1$ and 2 levels are found experimentally to be scattered into angular ranges up to ~20° and ~50°, respectively; while the trajectory surface-hopping calculation predicts that both $v' = 1$ and 2 levels are mainly scattered into angular ranges within ~10°, as shown in Fig. 3a, b. These are much larger scattering angular ranges than those when colliding with the spin-orbit ground $Ar^+(^2P_{3/2})$ ion as reported recently[27]. For $Ar^+(^2P_{3/2})$, the measured $N_2^+$ product is much more concentrated in the forward region (within 10°) compared with that of $Ar^+(^2P_{1/2})$ in this study, which has been reproduced well by the theoretical calculation[27]. This fact indicates that the much broader angular distribution for $Ar^+(^2P_{1/2})$ observed in the current experiment should not be due to any experimental broadening effects. The $N_2^+$ products with $v' > 3$ are mainly scattered in the backward direction, as shown in Fig. 1a. This could be attributed to head-on collisions, which mainly cause backward scattering with high vibrational excitation[28]. As the COM collision energy decreases, both experiments and theoretical calculations show that the charge-transfer products are gradually scattered into relatively larger angular ranges, as shown in Supplementary Fig. 2. A similar trend has also been noticed for the charge-transfer reaction between the spin-orbit ground $Ar^+(^2P_{3/2})$ ion and $N_2$[15,27].

The product rotational distributions in the $v' = 1$ and 2 levels and their correlations with the scattering angles calculated by the trajectory surface-hopping method at the collision energy of 1.58 eV are presented in Fig. 4. Those at the other three lower collision energies

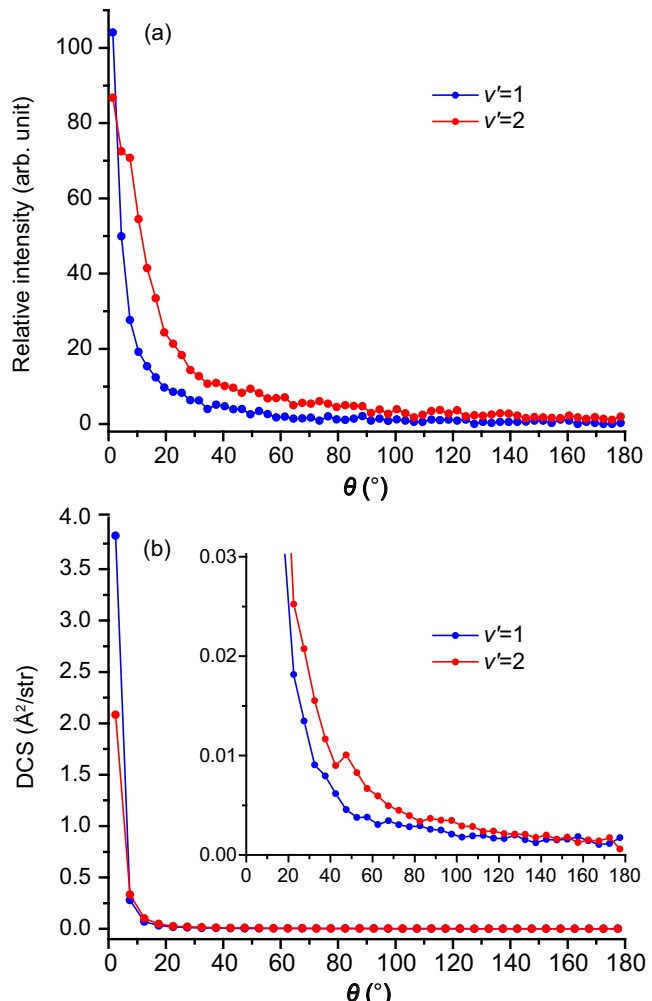

**Fig. 3 | Product angular distributions. a** Experimentally measured angular distributions of the $N_2^+$ product in the $v' = 1$ (blue) and 2 (red) levels at the COM collision energy of 1.58 eV for the charge-transfer reaction $Ar^+(^2P_{1/2}) + N_2 \rightarrow Ar + N_2^+$ $(v', J')$. **b** Calculated $N_2^+$ product angular distribution by the trajectory surface-hopping method, and the zoomed-in distribution is shown in the inset.

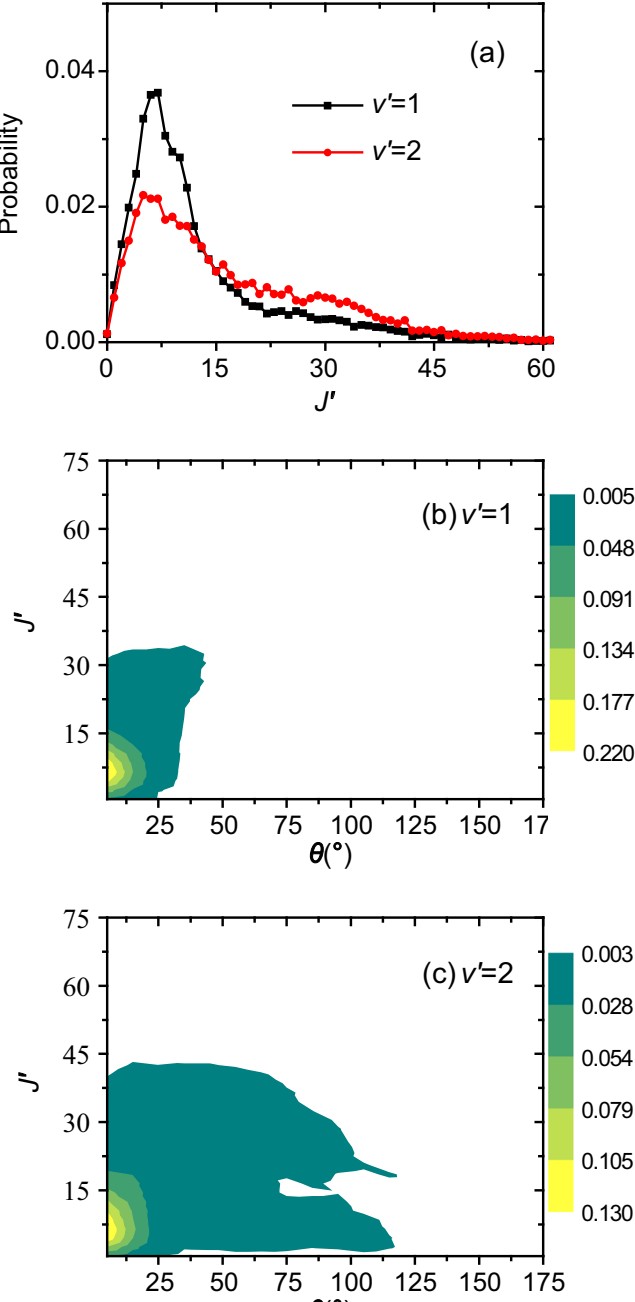

**Fig. 4 | Product rotational distributions and their correlations with the scattering angles. a** $N_2^+$ product rotational distributions in the $v' = 1$ (black) and 2 (red) levels for the charge-transfer reaction $Ar^+(^2P_{1/2}) + N_2 \rightarrow Ar + N_2^+(v', J')$ at the COM collision energy of 1.58 eV calculated by the trajectory surface-hopping method. The calculated correlation contour maps between the $N_2^+$ product rotational distribution and the scattering angle at 1.58 eV are presented in **b**, **c** for the $v' = 1$ and $v' = 2$ levels, respectively.

are shown in Supplementary Fig. 3. For the $v' = 1$ level, the rotational distribution strongly peaks at $J' = 6$ or 7, which is slightly higher than the peak at $J' = 3$ for the charge-transfer process between $Ar^+(^2P_{3/2})$ and $N_2$ at 1.588 eV as reported recently[27]. For the $v' = 2$ level, the rotational distribution is hotter than the $v' = 1$ level with a broad peak at $J' = \sim 5$. This is in sharp contrast to the charge-transfer process between the spin-orbit ground $Ar^+(^2P_{3/2})$ ion and $N_2$, for which the rotational distribution of the $v' = 2$ levels was calculated to peak at $J' = \sim 30$[27]. The higher rotational excitation in $v' = 1$ than that in $v' = 2$ makes the two product vibrational peaks closer to each other, thus more difficult to resolve from each other. This qualitatively agrees with the experimental observations. The $v' = 1$ and 2 vibrational levels were clearly resolved from each other in the forward scattering direction for the charge-transfer reaction between the spin-orbit ground $Ar^+(^2P_{3/2})$ ion and $N_2$ as reported recently[27], while they are only partially resolved here for that between the spin-orbit excited $Ar^+(^2P_{1/2})$ ion and $N_2$ as shown in Fig. 1.

The correlation contour map of the $v' = 1$ level also differs from that of the $v' = 2$ level, as shown in Fig. 4b, c. The colder rotational distribution of the $v' = 1$ level is predominantly scattered into the forward scattering region, and the much hotter rotational distribution of the $v' = 2$ level is scattered into relatively larger scattering angles. As the rotational excitation increases, the $v' = 2$ level products are

scattered into increasingly larger angles as shown in Fig. 4c. This qualitatively agrees with the current experimental observations. As shown in the scattering image in Fig. 1a, the $N_2^+$ product signal starts from the concentric ring corresponding to $v' = 2$ and gradually moves toward the concentric ring of $v' = 3$ as the scattering angle increases. This is also seen in the $N_2^+$ product speed distributions shown in Fig. 1b, where the $v' = 2$ peak gradually moves toward a lower speed (thus higher rotational excitation) as the scattering angle increases. A similar trend was observed for the $v' = 1$ level in the charge-transfer reaction between the spin-orbit ground $Ar^+(^2P_{3/2})$ ion and $N_2$[27]. As the collision

energy decreases, the rotational distribution in the $v' = 1$ vibrational level gradually peaks at higher rotational $J'$ levels, as shown in Supplementary Fig. 3. This is qualitatively consistent with the scattering images shown in Supplementary Fig. 1, where the strong signal in the forward direction gradually deviates from the concentric ring of $v' = 1$ as the collision energy decreases. Similar to the $v' = 1$ level in the charge-transfer reaction between the spin-orbit ground $Ar^+(^2P_{3/2})$ ion and $N_2$ as observed recently[27], the trend that higher rotational levels are scattered into larger scattering angles in the $v' = 2$ level as described above also fades away as the collision energy decreases. For example, the rotational excitation is almost independent of the scattering angle at the collision energy of 0.57 eV, as shown in Supplementary Fig. 3i. This qualitatively agrees with the experimental observations. The corresponding concentric rings overlap with the $v' = 2$ level well also at large scattering angles at the two collision energies of 0.83 and 0.57 eV, as shown in Supplementary Fig. 1b, c.

## Discussion

Comparing with $Ar^+(^2P_{3/2})$, the $N_2^+$ product from the $Ar^+(^2P_{1/2})$ reaction with $N_2$ has a higher $v' = 2$ population according to the trajectory surface-hopping calculation, increasing from no more than 10% for $Ar^+(^2P_{3/2})$ to ~42% for $Ar^+(^2P_{1/2})$, as shown in Fig. 2. Analysis of the trajectories revealed that the hopping mostly occurs near $R = 5$ Å, consistent with the long-range harpooning mechanism. A charge-transfer trajectory typically hops first to the $Ar^+(^2P_{3/2}) + N_2$ state, followed by a second hop to the product channel. This process can be rationalized by the energetics at large Ar-$N_2$ distances. As shown by the blue dash line in Fig. 5a, which corresponds to the energy of $N_2(v = 0)$ in the $Ar^+(^2P_{1/2}) + N_2$ channel, the inner turning point is about $r = 1.04$ Å, very close to the crossing seam (about $r = 1.02$ Å) between the $Ar^+(^2P_{1/2}) + N_2$ and $Ar^+(^2P_{3/2}) + N_2$ PESs. The $Ar^+(^2P_{3/2}) + N_2$ PES has another crossing seam with the product channel near $r = 1.06$ Å, which facilitates the final charge transfer. Compared with the $Ar^+(^2P_{3/2}) + N_2$ channel, for which the corresponding $N_2(v = 0)$ energy is marked in the same figure

by a red dashed line, the $Ar^+(^2P_{1/2})$ reactant affords higher vibrational excitation in the $N_2^+$ product. This is illustrated in Fig. 5b, where the distribution of the $r_{max}$ (the outer turning point in N−N vibration) is shown for the two spin-orbit states and the peak for $Ar^+(^2P_{3/2})$ is clearly smaller than the $Ar^+(^2P_{1/2})$ counterpart. Furthermore, the peak position for $Ar^+(^2P_{3/2})$ is near the outer turning point for $v' = 1$ of $N_2^+$, labeled by the middle black dotted line in Fig. 5a, consistent with the vibrational distribution reported in our earlier work[27]. For $Ar^+(^2P_{1/2})$, on the other hand, the peak position is between $v' = 1$ and $v' = 2$, marked by the two upper black dotted lines in the same figure. This is consistent with the increased $v' = 2$ population in Fig. 2. The different charge-transfer mechanism of $Ar^+(^2P_{1/2})$ spin-orbit state also explains the lack of the vibrational specificity observed for the $Ar^+(^2P_{3/2})$ channel.[27]

However, the agreement with the experimental observations for $Ar^+(^2P_{1/2}) + N_2$ reported here is not as good as for the $Ar^+(^2P_{3/2}) + N_2$ reaction reported in our earlier work[27]. The failure to quantitatively reproduce the experimental observations is presumably due to a number of factors such as the neglect of the coupling of the vibrational mode with other coordinates and errors in the empirical potentials and couplings. A full-dimensional diabatic potential energy matrix (DPEM) based on high-level ab initio calculations is needed to resolve these discrepancies.

The $v' = 1/v' = 2$ ratio for the $Ar^+(^2P_{1/2})$ DCS (Fig. 3) is smaller than that for $Ar^+(^2P_{3/2})$, which can be explained by the correlation maps in Fig. 5c, d between the impact parameter $b$ and scattering angle $\theta$ of the charge-transfer trajectories. Unlike $Ar^+(^2P_{3/2})$ where the $v' = 2$ angular distribution is much more isotropic than that for $v' = 1$, thanks to the vibration-specific mechanisms[27], the corresponding $v' = 2$ angular distribution for $Ar^+(^2P_{1/2})$ is also dominated by forward scattering, much like that for $v' = 1$. This can be attributed to the fact that both the $v' = 1$ and $v' = 2$ products in the $Ar^+(^2P_{1/2}) + N_2$ reaction are formed predominantly by large impact parameter collisions, via long-distance charge transfer and forward scattering.

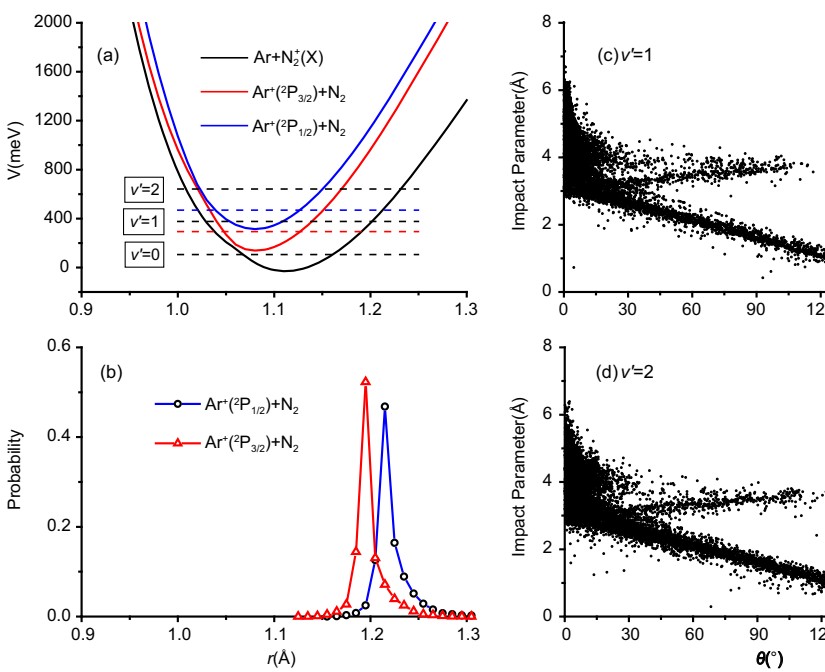

**Fig. 5 | Trajectory surface-hopping calculations. a** Adiabatic potential energy curves at $R = 7$ Å ($R$ is the distance between Ar and $N_2$ center of mass). The red and blue dashed lines are corresponding to $N_2(v = 0)$ level in the $Ar^+(^2P_{3/2}) + N_2$ and $Ar^+(^2P_{1/2}) + N_2$ channels, respectively. The three black dashed lines represent the $v' = 0, 1, 2$ vibrational levels of $N_2^+$ in the $Ar + N_2^+(X)$ channel, respectively. **b** Distribution of largest $r$ (the N−N distance) when the initial state is $Ar^+(^2P_{3/2}) + N_2$ at $E_c = 1.59$ eV or $Ar^+(^2P_{1/2}) + N_2$ at $E_c = 1.58$ eV; **c, d** Correlation plots between impact parameters and scattering angle ($\theta$) for $v' = 1$ and 2 of $N_2^+$.

The fact that both $N_2^+(v' = 1$ and 2) are formed through the same mechanism for the $Ar^+(^2P_{1/2})$ reaction also manifests in the product rotational distribution. Unlike $Ar^+(^2P_{3/2})$ where the rotational distributions of $v' = 1$ and $v' = 2$ peak at $J' = 3$ and $J' = 29$, respectively[27], for $Ar^+(^2P_{1/2})$, the rotational distributions for $v' = 1$ and $v' = 2$ show a similar peak at $J' = 7$ and $J' = 5$, respectively. This can again be understood by the glancing scattering with large impact parameters ($b > 3$ Å), which induces little rotational excitation.

In summary, we reported here a synergistic experimental and theoretical investigation on the charge-transfer dynamics between the spin-orbit excited $Ar^+(^2P_{1/2})$ and $N_2$, which has been seldom studied before. The product vibrational, rotational, and angular distributions are partially resolved by the high-resolution three-dimensional VMI imaging. Qualitatively, the trajectory surface-hopping calculations reproduced several main features of the experimental observations, and both experiment and theory clearly showed that the detailed microscopic charge-transfer mechanisms of the spin-orbit excited state $Ar^+(^2P_{1/2})$ are substantially different from that of the spin-orbit ground state $Ar^+(^2P_{3/2})$ as reported in our earlier study[27]. Quantitatively, however, several discrepancies were noticed between experiment and theory. The experiment observed a much higher population in the $N_2^+(v' = 2)$ level than that predicted by theory (Fig. 2). The DCSs deduced from the scattering images clearly showed that the $v' = 2$ level is scattered into much larger angles than the $v' = 1$ level, while the trajectory surface-hopping calculation predicted that the angular distributions of the $v' = 1$ and 2 levels are only slightly different (Fig. 3), as they are both formed predominantly by the long-range harpooning mechanism (Fig. 5). Furthermore, the experiment observed products into much larger scattering angles (20–50°) than that predicted by theory (within 10°). Such discrepancies could be caused by the empirical PESs and coupling strengths used in the current study and the neglect of the coupling of the N−N vibrational mode with other coordinates. A quantitative understanding of the microscopic charge-transfer dynamics between the spin-orbit excited $Ar^+(^2P_{1/2})$ and $N_2$ may require accurate PESs based on high-level ab initio calculations and dynamical calculations considering all necessary couplings.

## Methods
### Experimental
The ion-molecule crossed-beam setup used in this study with three-dimensional VMI and a photoionization-based quantum state selected pulsed ion beam source was described in detail previously[25–27]. The spin-orbit excited $Ar^+(^2P_{1/2})$ ion was produced by photoionizing a pulsed pure Ar beam at the UV wavelength of 372.765 nm, which photoionized Ar atoms through a (4 + 1) REMPI process and prepared $Ar^+$ ions in the spin-orbit excited $^2P_{1/2}$ state with a purity of ~95%[29]. The UV laser was generated by doubling the fundamental output of a dye laser (LiopTec, LIOPSTAR-E) pumped by a 10 Hz Nd:YAG laser (Beamtech, Nimma-900). The UV laser was focused into the photoionization region by a plano-convex lens with a focal length of 15 cm. The pulse energy of the UV laser was controlled to be ~7.5 mJ, so only about 200 $Ar^+$ ions were produced for each pulse, which could mitigate the space charge effect as much as possible, and at the same time maintain enough signal level for detection. The produced $Ar^+$ ions were then accelerated to ~120 eV and reached the crossing region within ~23 microseconds. Before arriving at the reaction center, the $Ar^+$ ions were focused and decelerated to the target kinetic energy (several eV) used for the collision experiment. A well-focused ion beam in the reaction center is pivotal for achieving scattering imaging with high energy resolution. We designed a tandem double Einzel lens system, which was more robust and could focus the ion beam better than the single Einzel lens setup[26,27]. The $Ar^+$ ion beam crossed at 90° in the reaction center of the VMI stack with the neutral pulsed supersonic $N_2$ beam (~40 K), which was produced by a general valve (Parker, Series

9). The exact velocities of the $Ar^+$ ion and $N_2$ beams were measured by the VMI setup. After crossing, high voltage pulses were switched on and the three components of the $N_2^+$ product velocity were measured by the three-dimensional VMI setup[25]. For each scattering image, ~100,000 product ions were collected to have enough statistics. According to the previous study, the COM collision energy spread in the forward direction is ~65 meV at $E_c = 1.58$ eV, and the intrinsic energy resolution of the VMI detection system should be better than 65 meV in the forward direction[27].

### Theoretical
The PESs used for the theoretical calculations are adapted from the 5 × 5 empirical diabatic potential energy matrix (DPEM) of Candori et al.[12]. To simplify the model, we only use three states in this work, $Ar^+(^2P_{3/2,1/2}) + N_2(X^1\sum_g^+)$, $Ar^+(^2P_{1/2,1/2}) + N_2(X^1\sum_g^+)$, and $Ar + N_2^+(X^2\sum_{g,\frac{1}{2}}^+)$. The details have been described in our previous work[27]. In addition, Morse functions of $N_2$ and $N_2^+$ are added to the corresponding PESs, with the explicit assumption that the vibrational degree of coordinate is decoupled with the other nuclear degrees of freedom.

The fewest switches with time uncertainty (FSTU) method[30] implemented in the ANT program[31] was used in the dynamics calculations. Nonadiabatic transitions were followed in the adiabatic representation using the stochastic decoherence (SD) scheme used with FSTU[32], while the grad$V$ prescription was used for all frustrated hops[33]. The initial state of $N_2$ was specified by $v = 0$, $J = 0$ with the initial separation between the collision partners at 8 Å. A trajectory is terminated when the two are separated by 10 Å, where the vibrational and rotational quantum numbers of the $N_2^+$ products were then determined, as described in our earlier work[27]. The impact parameter ($b$) was sampled from a uniformly distributed random number $\zeta \in [0,1]$, according to $b = b_{max}\zeta^{1/2}$, where $b_{max}$ equals the initial reactant separation between the collision partners (8.0 Å).

## Data availability
Data are provided in this paper and can be downloaded at https://doi.org/10.6084/m9.figshare.24523192[34]. Source data are provided in this paper.

## Code availability
The ANT program is from Donald G. Truhlar and can be downloaded from https://comp.chem.umn.edu/ant/[31].

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

## Acknowledgements

This work was funded by the National Natural Science Foundation of China (No. 22373107 to H. Gao), Beijing Municipal Natural Science Foundation (No.1222033 to H. Gao), and Air Force Office of Scientific Research (FA9550-22-1-0350 to H. Guo). H. Gao is also supported by the K. C. Wong Education Foundation and the Innovation Capability Support Program of Shaanxi Province (2023-CX-TD-49). The computation was performed at the Center for Advanced Research Computing (CARC) at UNM. We are very grateful to David Cappelletti for sharing with us the PESs.

## Author contributions

The experiments were conceived and supervised by Hong Gao and carried out by Guodong Zhang. Theoretical calculations were conceived by Hua Guo and performed by Dandan Lu. The paper was written by Hong Gao, with the theoretical sections contributed by Hua Guo and Dandan Lu. All authors contributed to discussions about the results and manuscript.

## Competing interests

The authors declare no competing interests.
