## [Peer Review File · Nature Communications]

Imaging the state-to-state charge-transfer dynamics between the spin-orbit excited Ar+(2P_{1/2}) ion and N₂Reviewers' Comments:

Reviewer #1:

Remarks to the Author:

In this manuscript Gao and coworkers report the reaction dynamics studies of the $\text{Ar}^+ + \text{N}_2$ charge transfer reaction with the argon ion prepared selectively in the upper $j=1/2$ spin-orbit state of the ground electronic state. This selectivity, which the authors have already successfully applied to the charge transfer reaction of the lower $j=3/2$ Ar^+ spin orbit state, is the main experimental novelty of this work and provides a much better insight into these dynamics than what was previously possible. The present results are compared point by point with the previously published result for the other Ar^+ spin-orbit ground state. In contrast to the $j=3/2$ result, a full product vibrational resolution was not achieved in the present work, which is explained by a more similar rotational excitation of the two main vibrational levels of the N_2^+ product. The experimental results are compared with detailed theoretical surface hopping calculations, which show overall a good agreement, but fail to reproduce specific details quantitatively, such as the broader angular distribution or the relative ratio of the $v=1$ and 2 levels. However, it is the strength of the present work that such a detailed comparison is possible at all and it will most certainly stimulate more accurate calculations in the future. In summary I consider this work a major step forward in understanding the dynamics of fundamental charge transfer reactions and therefore recommend it for publication.

Specific points that should be addressed in a revised manuscript:

- * In Figure 1, are the energy values for the N_2^+ vibrational levels calculated in the harmonic approximation or do they include anharmonic corrections? This should be stated in the caption.
- * The collision energies are always quoted with the least significant digit being 1meV. I doubt that the energy resolution of the experiment is good enough to have such well defined collision energies. If not then the energies should be quoted with a lower number of digits.
- * The comparison in Figure 2 is difficult to read. At first glance I thought there is a very good agreement between theory and experiment, before I understood which bars represent $v=1$ and which $v=2$. Maybe the authors can think of a better way to represent the data.
- * On page 7 the authors state that "This may be due to the inaccuracies of the PESs used in the calculation.", which I find a little too simplified. Later on page 12 they refer to the need for better potential energy matrices, which is more specific and probably better describes the problem of having to treat several coupled electronic surfaces. The last sentence of the article then just refers again to "accurate PESs". I think this should be reworded.
- * I did not notice that an experimental resolution is included in the direct comparison with the theoretical distributions, e.g. in Figure 3? Could an experimental broadening of the angular distribution lead to a more favorable comparison with the theory? There should be a comment about this in the caption or the main text.

Reviewer #2:

Remarks to the Author:

Report on NCOMMS-23-32193 by Gao et al.

In this manuscript, the authors investigated the state-to-state differential cross sections for the charge-transfer reaction $\text{Ar}^+(2P_{1/2}) + \text{N}_2 \rightarrow \text{Ar} + \text{N}_2^+$, using the ion-molecule crossed beam machine with three-dimensional VMI technique. The main features of the product vibrational, rotational and angular distributions are qualitatively consistent with those from the trajectory surface-hopping

calculations. Both the experiment and the theory show that the microscopic charge-transfer mechanisms of $\text{Ar}^+(2P_{1/2})$ are quite different from the previously reported spin-orbit ground state $\text{Ar}^+(2P_{3/2})$.

The manuscript is well organized and the conclusions are supported by the data.

I have the following questions for the authors.

(1) In the current reaction of $\text{Ar}^+(2P_{1/2}) + \text{N}_2$, some discrepancies were noticed between the trajectory surface-hopping calculations and the scattering experiments. However, in the spin-orbit ground reaction $\text{Ar}^+(2P_{3/2}) + \text{N}_2$, the results from the experiment and the theory match quite well. What are the main reasons causing the discrepancy?

(2) In fig. 2, what's the error from the experiment? I think it will be helpful to add the error bars.

(3) In S2 part, when estimating the N_2^+ product vibrational distributions, the peak widths are about the same. Is there any reason for this? From the raw images, the excitation of product rotational states varies greatly with the vibrational levels and the angles. Then the vibrational widths may vary largely.

(4) On line 328 of the experimental section, it says that only about 200 Ar^+ ions were produced for each laser pulse. How did you estimate this number? What's the speed ratio of Ar^+ beam? At this ion density, how long would it take to collect ~ 100000 product ions?

In summary, this is a beautiful job in the study of ion-molecule reactions. I recommend the publication of this paper.

Reviewer #3:

Remarks to the Author:

The manuscript investigates a benchmark reaction in ion molecule reaction dynamics, that is the charge transfer reaction between Ar^+ and molecular nitrogen N_2 . The present manuscript presents study in line with a previous publication by the authors on the reaction of the Ar^+ in its $P_{3/2}$ state with N_2 . The reaction is characterized by a resonant channel forming N_2 in its first vibrationally excited state rather than in its ground vibrational state. The present manuscript investigates the reaction of Ar^+ in its $P_{1/2}$ state which is formed by laser ionization to generate a pure beam of Ar^+ ions in a state-selected fashion. The experimental and theoretical methodology follows the recent publication in Nature Chemistry. To be able to extend the experiments and the theory to a second state might seem trivial at the first glance but it is far from this. Each state requires a carefully chosen ionization scheme for the experiment and the ratio of $P_{3/2}$ to $P_{1/2}$ disfavors the current experiment compared to the $P_{3/2}$ state. In case of theory, one cannot simply "copy" the $P_{3/2}$ PES. Therefore, I highly appreciate the experimental and theoretical effort of the authors regarding their study and recommend it for publication in Nature Communications after the authors address the minor comments below.

I am curious if the authors can "artificially" generate the scattering distributions of a statistical Ar^+ ion beam from their theoretical data and if that than would agree with the experimental data of Wester and co-workers who assumed their Ar^+ state population to be statistical.

In the discussion of the integrated angular distributions the authors comment on the deviation of experiment and theory regarding the large angle scattering. They mention "inaccuracies in the PES" which they explain in more detail in the discussion. It might be helpful to refer the reader to this later discussion because I was asking myself at this point what these inaccuracies are when reading the paragraph.

Figure 1

What are the units of the 2D histogram? Is normalized to the highest intensity or are this absolute counts. And how significant are the structures seen in the velocity distributions. It would be helpful for the authors to give an uncertainty. In the text they refer to "partially resolved $v'=1$ and $v'=2$ ".

II. 144/145 and II 152-155

How does the assignment of internal excitation to rotational degrees of freedom work? It reads as if the assignment of "rotational" excitation is based on the experiment alone which is confusing for the reader because that is not the case and clearly stated later in the manuscript. However, the authors might want to clarify this in this paragraph to avoid confusion. Further, the statements in lines 144/145 and 152-155 read somewhat contradictory.

II. 284/285

The sentence is confusing. Maybe the authors want to remove the first part or rephrase.

II.296/297

The paragraph repeats the content from the paragraph before or I did not get the intended meaning. Please clarify.

Reviewer #4:

Remarks to the Author:

This manuscript presents a detailed experimental and theoretical study on the $\text{Ar}^+(^2P_{3/2}) + \text{N}_2 \rightarrow \text{Ar} + \text{N}_2^+$ charge transfer reaction. For the first time, the initial Ar ion is formed in the pure $^2P_{3/2}$ excited spin-orbit state, and the differential cross section is measured for the N_2^+ channel, for several collision energies, under single collision conditions. This allowed to determine the final vibrational state of the N_2^+ products. The final distribution cannot be resolved because of the congestion. Theoretical simulations, using quasi-classical surface hopping method in a model of three electronic states, are also presented showing a reasonable qualitative agreement with the measured results. The differences are attributed to inaccuracies in the potential energy surfaces used. The results are also compared with previous experiments, performed in the same laboratory for the ground $\text{Ar}^+(^2P_{1/2})$ spin-orbit state. The results are interesting and the experimental development is a novelty. I find, however, that some explanations are needed, specially in the theoretical side, to give a complete picture of the charge transfer process, specially when comparing with the initial spin-orbit state results, as listed below:

1) the theoretical results show a different final vibrational distribution, peaked at $v'=1$ instead of $v'=2$ found in the experiments.

Since the theoretical study is done classically, a description on how "quantized" rovibrational state are assigned. Also, a comment on how initial conditions are selected are desirable.

2) Concerning this problem, why not using a quantum method in which some of the authors have a deep experience? Dealing with final state cross sections a full quantum method would be more adapted to better understand the detailed experimental results. Also, the electronic transitions would be better described.

3) It is discussed the a "vibrational-specific mechanism" explains the results for the ground spin-orbit state. Why not for the excited $\text{Ar}^+(^2P_{3/2})$ one?. This specificity occurring at long distance is lost when using classical mechanics. A quantum method should then be used

Updates on theoretical results

During the revision, we discovered a small error in calculating the product state distributions and the results are updated in Figures 2 - 5. The changes are very small and the conclusions are not altered.

Reviewer #1:

Comment 1: In Figure 1, are the energy values for the N_2^+ vibrational levels calculated in the harmonic approximation or do they include anharmonic corrections? This should be stated in the caption.

Author reply: The anharmonic terms are considered in the calculation of the N_2^+ vibrational levels. We have added this to the figure caption.

Comment 2: The collision energies are always quoted with the least significant digit being 1 meV. I doubt that the energy resolution of the experiment is good enough to have such well-defined collision energies. If not, then the energies should be quoted with a lower number of digits.

Author reply: Thanks for this suggestion. Indeed, the resolution of the kinetic energy measurement cannot reach the level of 1 meV. Based on our imaging resolution, each pixel of the camera corresponds to a speed of ~ 21 m/s for the Ar^+ ion. In our experiment, the maximum Ar^+ ion velocity is ~ 4200 m/s, each pixel represents an ion beam kinetic energy interval of ~ 36 meV, which corresponds to a collision energy of ~ 16 meV in the COM frame. Thus, we now quote the experimental collision energies to the second digit (~ 10 meV), which should be more reasonable. The theoretical values are accurate with the stated significant figures.

Comment 3: The comparison in Figure 2 is difficult to read. At first glance I thought there is a very good agreement between theory and experiment, before I understood which bars represent $v = 1$ and which $v = 2$. Maybe the authors can think of a better way to represent the data.

Author reply: Thanks for reminding us on this issue! It might be due to the color scheme we used, which could be a bit misleading. We now changed the color scheme to that used in our previous Nature Chemistry paper. We think it looks much better now.

Comment 4: On page 7 the authors state that "This may be due to the inaccuracies of the PESs used in the calculation.", which I find a little too simplified. Later on page 12 they refer to the need for better potential energy matrices, which is more specific and probably better describes the problem of having to treat several coupled electronic surfaces. The last sentence of the article then just refers again to "accurate PESs". I think this should be reworded.

Author reply: We have now removed the sentence "This may be due to the inaccuracies of the PESs used in the calculation.". We have also reworded the last two sentences at the end the

main text in the revised manuscript.

Comment 5: I did not notice that an experimental resolution is included in the direct comparison with the theoretical distributions, e.g. in Figure 3? Could an experimental broadening of the angular distribution lead to a more favorable comparison with the theory? There should be a comment about this in the caption or the main text.

Author reply: Thanks for the good question! The translational energy resolution of the setup was described in our previous Nature Chemistry paper. A sentence for describing this has been added to the experimental section of the current manuscript. The intrinsic angular resolution of the setup is difficult to characterize. However, we can judge this from the charge-transfer images between the spin-orbit ground $\text{Ar}^+(\text{}^2\text{P}_{3/2})$ ion and N_2 (reported in our previous Nature Chemistry paper), which were undertaken at the same experimental conditions with the current experiment. For $\text{Ar}^+(\text{}^2\text{P}_{3/2})$, the measured product N_2^+ is much more concentrated in the forward region (within 10 degrees) compared with that of $\text{Ar}^+(\text{}^2\text{P}_{1/2})$ in the present study, which has been reproduced well by the theoretical calculation (see Figure 3 in our recent Nature Chemistry paper). This fact indicates that the much broader angular distribution for $\text{Ar}^+(\text{}^2\text{P}_{1/2})$ observed in the current experiment cannot be due to experimental broadening effect. We have added a comment about this in the revised manuscript.

Reviewer #2:

Comment 1: In the current reaction of $\text{Ar}^+(\text{}^2\text{P}_{1/2}) + \text{N}_2$, some discrepancies were noticed between the trajectory surface-hopping calculations and the scattering experiments. However, in the spin-orbit ground reaction $\text{Ar}^+(\text{}^2\text{P}_{3/2}) + \text{N}_2$, the results from the experiment and the theory match quite well. What are the main reasons causing the discrepancy?

Author reply: We have tried our best to keep the experimental conditions for the $\text{Ar}^+(\text{}^2\text{P}_{3/2})$ and $\text{Ar}^+(\text{}^2\text{P}_{1/2})$ experiments the same, and the theoretical calculations were performed by using the same PESs and same method, thus we are not exactly sure the main reasons of the discrepancy. Inaccuracy of the PESs could be one of the reasons. Indeed, the potentials are expressed as a sum of a two-dimensional (R, θ) potential and a morse potential of r . Hence the couplings of the vibrational mode with other coordinates are not accounted for. Furthermore, the empirical nature of the potentials may also cause errors. Finally, we note that the charge transfer involving $\text{Ar}^+(\text{}^2\text{P}_{1/2})$ needs to undergo two effective nonadiabatic transitions, as opposed to one for $\text{Ar}^+(\text{}^2\text{P}_{3/2})$. This might also cause less perfect agreement with experiment.

Comment 2: In fig. 2, what's the error from the experiment? I think it will be helpful to add the error bars.

Author reply: Thanks for the suggestion! As we have mentioned in the SM, the N_2^+ product vibrational levels are not completely resolved, and the products show complicated rotational

and angular distributions, thus it is not possible to make an accurate measurement on the product vibrational branching ratios and the uncertainties. The only thing we know is the fitting error to the experimental data, which is much smaller than 1%, thus almost not visible under the scale of Figure 2. We added the sentence “The fitting error is much smaller than 1%, thus not shown in Figure 2.” to the last paragraph on page 6.

Comment 3: In S2 part, when estimating the N_2^+ product vibrational distributions, the peak widths are about the same. Is there any reason for this? From the raw images, the excitation of product rotational states varies greatly with the vibrational levels and the angles. Then the vibrational widths may vary largely.

Author reply: This is true! We have discussed this in SM about this. The N_2^+ product vibrational levels are not completely resolved, and the products show complicated rotational and angular distributions, thus it is not possible at all to make an accurate measurement on the product vibrational branching ratios. Only a rough estimation was made by the fitting process as described in SM. In the fitting process, we divided the scattering images into three angular ranges according to the angular dependences of the vibrational and rotational excitations of N_2^+ . Considering that the interval change of vibrational energy levels is small, and the products at different vibrational energy levels may all have high rotational energy, the peak widths are adjusted to be about the same for all vibrational levels. The peak positions are adjusted slightly to have the best agreement with the experimental curves.

Comment 4: On line 328 of the experimental section, it says that only about 200 Ar^+ ions were produced for each laser pulse. How did you estimate this number? What's the speed ratio of Ar^+ beam? At this ion density, how long would it take to collect ~ 100000 product ions?

Author reply: The number of Ar^+ ions in the beam was estimated by using the conventional 2D imaging scheme, which can count the number of ions hitting on the detector for each pulse. The energy spread of the Ar^+ beam is ~ 278 meV at the total kinetic energy of 3.71 eV. We have discussed the energy resolution of the experiment in detail in the previous Nature Chemistry paper, and also mentioned it in the current manuscript. At this ion density, it took ~ 20 hours to collect ~ 100000 product ions.

Reviewer #3:

Comment 1: I am curious if the authors can "artificially" generate the scattering distributions of a statistical Ar^+ ion beam from their theoretical data and if that than would agree with the experimental data of Wester and co-workers who assumed their Ar^+ state population to be statistical.

Author reply: We have combined the experimental scattering images of $Ar^+(^2P_{3/2})$ and $Ar^+(^2P_{1/2})$ as shown in the following figure. The images are weighted by the integral cross sections of

$\text{Ar}^+(\text{}^2\text{P}_{3/2})$ and $\text{Ar}^+(\text{}^2\text{P}_{1/2})$, which are ~ 14.5 and 4.5 \AA^2 , respectively, and the statistical ratio of 2:1. The combined image does look more like that of $\text{Ar}^+(\text{}^2\text{P}_{3/2})$ due to the much less contribution of the $\text{Ar}^+(\text{}^2\text{P}_{1/2})$ ion. The combined image appears to be similar to that of Wester's result in the forward direction, but the resolution seems to be higher as the $v=1$ and $v=2$ rings are partially resolved.

Comment 2: In the discussion of the integrated angular distributions the authors comment on the deviation of experiment and theory regarding the large angle scattering. They mention "inaccuracies in the PES" which they explain in more detail in the discussion. It might be helpful to refer the reader to this later discussion because I was asking myself at this point what these inaccuracies are when reading the paragraph.

Author reply: Thanks! To avoid any confusions here, we have deleted the sentence "This may be due to the inaccuracies of the PESs used in the calculation." in the revised manuscript.

Comment 3: Figure 1. What are the units of the 2D histogram? Is normalized to the highest intensity or are this absolute counts. And how significant are the structures seen in the velocity distributions. It would be helpful for the authors to give an uncertainty. In the text they refer to "partially resolved $v' = 1$ and $v' = 2$ ".

Author reply: The units of the 2D histogram are absolute counts. "The colour bar represents the absolute product ion count." has been added to the caption of Figure 1. For the image at each COM collision energy, it took several days to accumulate enough ion count (we tried to keep the experimental conditions the same everyday), and we finally added them up to make one image. In principle, at each collision energy, we only have one image, we thus do not have an uncertainty to the speed distribution spectra. However, the overall peak shapes, including the shoulder structure assigned as $v' = 1$ are very well reproducible. We had an image at slightly lower collision energy of 1.428 eV as shown in the figure below. It is almost the same with that at 1.577 eV, thus we did not present this in the paper. It shows that the overall peak profiles are repeatable. We have changed "shoulder" in line 126 to "reproducible shoulder". We hope this answered the questions of the reviewer.

Comment 4: ll.144/145 and ll.152-155 How does the assignment of internal excitation to rotational degrees of freedom work? It reads as if the assignment of "rotational" excitation is based on the experiment alone which is confusing for the reader because that is not the case and clearly stated later in the manuscript. However, the authors might want to clarify this in this paragraph to avoid confusion. Further, the statements in lines 144/145 and 152-155 read somewhat contradictory.

Author reply: As shown in Figure 1(b), the main peak gradually moves toward the lower speed direction (indicating larger internal excitation) at larger scattering angle; on the other hand, the peak position does not go beyond $v'=3$ (see the speed distributions in the angular ranges 15-30° and 30-50°), thus it cannot be due to vibrational excitation to $v'=3$, and can only be attributed to rotational excitation. We have revised lines 144-145 to make this clear. Although the internal excitation can be assigned to rotational degree of freedom, but clear resolution for each individual rotational and vibrational levels are still not possible, thus a definitive determination of the vibrational population is not feasible.

Comment 5: ll.284/285 The sentence is confusing. Maybe the authors want to remove the first part or rephrase.

Author reply: The first part is now removed.

Comment 6: ll.296/297 The paragraph repeats the content from the paragraph before or I did not get the intended meaning. Please clarify.

Author reply: This paragraph is meant to illustrate that the charge transfer mechanism for $\text{Ar}^+(^2P_{1/2})$ is different from that of $\text{Ar}^+(^2P_{3/2})$. Specifically, the extent of rotational excitation in both the $\text{N}_2^+(v=1)$ and $\text{N}_2^+(v=2)$ channels is quite similar for $\text{Ar}^+(^2P_{1/2})$, while that is quite different for $\text{Ar}^+(^2P_{3/2})$, due to the vibrational specific mechanisms discussed in our previous work (Ref. 27).

Reviewer #4:

Comment 1: The theoretical results show a different final vibrational distribution, peaked at $v' = 1$ instead of $v' = 2$ found in the experiments. Since the theoretical study is done classically, a description on how "quantized" rovibrational state are assigned. Also, a comment on how initial conditions are selected are desirable.

Author reply: The initial state was fixed at $v=0$, and $j=0$ and the quantization of the final rovibrational states was described in our previous work (Ref. 27), as clearly stated in the Materials and Methods section. No change has been made.

Comment 2: Concerning this problem, why not using a quantum method in which some of the authors have a deep experience? Dealing with final state cross sections a full quantum method would be more adapted to better understand the detailed experimental results. Also, the electronic transitions would be better described.

Author reply: This is a good question. A quantum dynamical treatment would indeed be very interesting, but challenging. For this system with three heavy atoms, a classical description of the nuclear dynamics is often considered acceptable. However, we do expect quantum treatment will be more reliable in treating nonadiabatic transitions and zero-point energy. We have plans to carry out such calculations in the future.

Comment 3: It is discussed the a "vibrational-specific mechanism" explains the results for the ground spin-orbit state. Why not for the excited $\text{Ar}^+(^2P_{3/2})$ one?. This specificity occurring at long distance is lost when using classical mechanics. A quantum method should then be used.

Author reply: The vibrational-specific mechanism for the ground state $\text{Ar}^+(^2P_{3/2})$ in our previous work is because the crossing point between the ground state and product state is $\sim 1.05 \text{ \AA}$ (see Figure 5), which is close the inner turning point of $\text{N}_2(v=0)$, also close the turning point of $\text{N}_2^+(v=1)$. As a result, it favors a long-distance resonant harpooning mechanism, while this is not the case for $\text{N}_2^+(v=2)$. For the excited state $\text{Ar}^+(^2P_{1/2})$, the crossing point between initial state and product state is $\sim 1.02 \text{ \AA}$. As a result, the effect of the turning point is small, and the difference between the $v=1$ and $v=2$ product channels is also small. We have added the following sentences in page 12 to make it clearer: "The different charge-transfer mechanism of $\text{Ar}^+(^2P_{1/2})$ spin-orbit state also explains the lack of the vibrational-specificity observed for the $\text{Ar}^+(^2P_{3/2})$ channel.²⁷".

Reviewers' Comments:

Reviewer #1:

Remarks to the Author:

The authors have provided detailed and suitable answers to my questions and I recommend the revised version for publication in Nature Communications.

Reviewer #2:

Remarks to the Author:

The authors revised the paper properly. I recommend the acceptance of this paper.

Reviewer #3:

Remarks to the Author:

The authors provided adequate replies to my remarks and made the respective changes in the manuscript. I have no further comments and support publication of the manuscript in Nature Communications.

Reviewer #4:

Remarks to the Author:

The authors have answered satisfactorily the questions or comments made by the four referees and it is now ready for publication